# Biomechanical Analysis of Silk as a Tendon or Ligament Replacement

**DOI:** 10.3390/polym17223052

**Published:** 2025-11-18

**Authors:** Caleb Wagner, Colin McCloskey, Kaitlin Williams, Katherine Teixeira, Benjamin D. Brooks, Amanda E. Brooks

**Affiliations:** 1Department of Biomedical Sciences, College of Osteopathic Medicine, Rocky Vista University–Southern Utah Campus, Ivins, UT 84738, USA; caleb.wagner@ut.rvu.edu (C.W.); abrooks@rvu.edu (A.E.B.); 2Department of Aerospace Medicine, Robins Air Force Base, Houston County, GA 31098, USA; 3Department of Orthopedic Surgery, Good Samaritan Regional Medical Center, Corvallis, OR 97330, USA; 4Department of Orthopedic Surgery, University of Pittsburgh Medical Center, Pittsburgh, PA 15213, USA

**Keywords:** tendon and ligament injuries, tissue engineering, orthopedic surgery, grafting options, tendon grafts, ligament grafts, spider silk

## Abstract

Advances in silk-based tissue-engineered constructs offer promising opportunities to improve tendon and ligament repair by increasing graft availability and enhancing patient outcomes. The rising demand for tendon and ligament reconstruction highlights the need for biomaterials that address limitations of autografts and allografts, including donor-site morbidity, limited supply, and immune rejection risks. Silk-based scaffolds leverage their tunable biomechanical properties—such as Young’s modulus, ultimate tensile strength, and strain to failure—to closely mimic native tendon and ligament function. This review synthesizes current literature on silk-derived grafts, summarizing their mechanical performance, fabrication strategies, and translational potential. Emphasis is placed on spider silk, which demonstrates exceptional tensile strength, elasticity, and biocompatibility, making it a strong candidate for next-generation scaffolds. Remaining challenges include optimizing in vivo degradation rates, enhancing tendon-to-bone (enthesis) integration, developing tunable structural and biochemical features, improving manufacturability, and validating clinical efficacy through standardized testing and robust clinical trials. Additional limitations to the application of silk as a biomaterial scaffold include high production costs, challenges associated with controlled spinning and processing, and the current lack of scalable manufacturing methods. Continued innovation and rigorous preclinical and clinical evaluation will be critical to realizing silk’s potential in advancing tendon and ligament repair and improving long-term functional outcomes.

## 1. Introduction

Tendon and ligament (T/L) injuries span many conditions, from minor sprains to complete tears, and are a significant and growing health concern, considerably impacting affected individuals and the broader healthcare system [1]. Tendon and ligament injuries are responsible for ~1/3 of all musculoskeletal consultations, with an annual incidence of four million worldwide. In the United States alone, the annual surgeries related to these injuries exceed 300,000 [2] (Figure 1) These figures highlight the growing number of T/L injuries, the possible complications that may arise from these injuries, and the extensive rehabilitation required to recover. Past research on T/L injuries provided valuable insights into T/L biological and mechanical properties. This knowledge has translated into advances in both surgical and conservative treatments for T/L injuries [2]. Despite these advances, T/L injuries remain a persistent clinical challenge [1]. The broader implications of these injuries manifest as decreased work productivity, extended physical therapy sessions, and the emotional challenges of managing reduced mobility and/or chronic discomfort [1].

Tendons are fibrous tissues that connect muscles to bones, enabling movement and providing stability. Ligaments, conversely, are fibrous tissues that connect bones to other bones, providing stability to joints and ensuring that they move within their physiological limits. Overexertion or age-related wear often makes T/L vulnerable to damage. Once injured, the healing process of T/L is notoriously slow; moreover, patients rarely regain the complete structural robustness and mechanical strength of their undamaged T/L [1]. This inherent limitation in T/L recovery not only presents therapeutic conundrums for clinicians but also places prolonged physical and emotional strain on patients, often manifesting as persistent discomfort, limited mobility, or the necessity for ongoing treatments [2].

Due to their similarities in biomechanical properties, porcine tissues have been an increasingly popular graft option during reconstructive T/L procedures to serve as an effective biological scaffold [3]. Additionally, porcine grafts facilitate graft vascularization and cell repopulation, which are essential for successful long-term integration, as observed in anterior cruciate ligament (ACL) and rotator cuff tendon reconstructions [3]. Porcine xenografts also eliminate the need for two-stage procedures in appropriate patient populations compared to auto- and allograft harvesting, reducing donor site morbidity [4]. Lastly, advancements in tissue engineering have minimized concerns regarding immunologic reactions through the decellularization and sterilization of xenografts [5].

While still an emerging practice, preliminary successes using porcine-derived T/L xenografts suggest a promising, biocompatible option when performing T/L reconstruction in select patients [6]. Although porcine-derived grafts have certain benefits, engineered tissue constructs could still be superior when evaluating donor tissue availability and graft quality. The advantages of biochemically produced synthetic scaffolds include the potential for exact control of tissue structure and characteristics, while eliminating dependence on animal-derived donor tissue [7].

## 2. Anatomy of Tendons and Ligaments

Tendons and ligaments have high resistance loads applied to them, which directly correlates to their structure (Figure 2). Composed of densely packed collagen fibers, the structure of tendons and ligaments allows for high-resistance loads. While tendons facilitate unidirectional tensile strength due to the parallel arrangement of collagen fibrils, ligaments allow for multi-axial flexibility and strength due to the interwoven, multi-directional collagen fibril pattern [7]. Approximately 65–80% collagen fibers and 1–2% elastin fibers are in tendons, and the collagenous matrix is attributed to the biochemical properties required for joint stabilization and functionality [8]. Additionally, surrounding proteoglycans, glycosaminoglycans, and glycoproteins result in a high level of water-binding capacity [8].

### 2.1. Anatomy of Tendons

Tendons connect to bones at the osteotendinous junction or enthesis, and tendons meet muscles at the myotendinous junction. These intricate junctions have meticulously evolved. The enthesis is indispensable in alleviating stress concentrations at the interface of hard and soft tissues, thereby protecting against collagen fiber abnormalities such as bending, shearing, and breakage [3,4]. Two distinct types of entheses exist: fibrous and fibrocartilaginous. In the case of the fibrous enthesis, the tendon or ligament directly connects to the bone. Conversely, the fibrocartilaginous enthesis is characterized by a graded transitional interface consisting of zones of uncalcified fibrocartilage, calcified fibrocartilage, and bone [4]. Tendons are susceptible to injury and degeneration, particularly at high-friction and stress-concentration sites. The enthesis is a common location that requires surgical repair techniques, as repeated microtraumas can lead to painful conditions such as insertional tendinopathies or tears at the tendon-bone interface.

### 2.2. Tendon Injury

Specific tendons have been observed to be more susceptible to damage than others. The rotator cuff, Achilles tendon, tibialis posterior, and patellar tendons are noted for their vulnerability [9]. This increased vulnerability is often linked to the unique biomechanical demands on these tendons and their specific anatomical positions. Consequently, these tendons are a focal point in preventive strategies and treatment protocols. Understanding their propensity for injury is crucial for developing practical approaches to tendon health, replacement suitability, and post-surgical care. Various factors can lead to tendon rupture, including a high-energy overload injury mechanism, overuse conditions causing tendinitis or tendinosis, or intrinsic tissue degeneration. The physiologic mechanical strain on tendons can be illustrated by a tendon stress–strain curve, which depicts the distinct phases of tendon deformation in response to applied stress (Figure 3). Mechanical strain that exceeds the physiologic range results in tendon fiber damage, which can ultimately progress to rupture. Tendon tear patterns and recovery potential vary depending on anatomic location, vascular supply, and local environment (e.g., intrasynovial vs. extrasynovial) [10]. Certain tendon ruptures have been shown to correlate with vascular supply, such as Achilles tendon tears, which virtually always occur in the same position of the watershed area, approximately 2–4 cm above the calcaneal insertion [11]. With the posterior tibialis tendons, rupture most commonly occurs in the hypovascular area of the tendon located at the posteromedial extent of the medial malleolus [12]. See Table 1 for common tendon rupture locations.

However, other tendon tear patterns may depend more on how the anatomic location and weakness contribute to the frequency of tears, such as the musculotendinous junction. In an analysis of 360 shoulders, Kim et al. found that the most common anatomic location of degenerative rotator cuff tears is approximately 15 mm posterior to the biceps tendon, and that most degenerative tears initiate near the junction of the supraspinatus and infraspinatus tendons [13].

Lastly, additional research studies are needed to assess the rupture patterns of other tendons. Tendons such as the patellar tendon have multiple locations of tear sites, and continued data is needed to assess which specific site is more susceptible. Most commonly, there is an avulsion from the inferior pole of the patella at the tendon-bone interface; other tear patterns include a mid-substance tear and a distal avulsion off the tibial tubercle [14].

### 2.3. Anatomy of Ligaments

Ligaments comprise dense connective tissue, predominantly type I collagen fibers, constituting about 70–80% of their dry weight [15]. Collagen fibrils are organized with bundles aligned along the long axis of the ligament, displaying an underlying “waviness” or crimp configuration throughout their length. This crimped organization plays a crucial biomechanical role in the ligament’s loading response, with increased loading resulting in areas of the ligament uncrimping, allowing the tissue to elongate without sustaining damage. The characteristic crimp pattern is essential for joints that experience varying loads, enabling ligaments to accommodate tensile forces while maintaining structural integrity. The collagen fibers’ crimped alignment provides the mechanical adaptability and protective elongation capacity essential to ligament function during joint motion [15]. Elastin fibers are interconnected among the collagen, adding essential elasticity that allows ligaments to stretch and recoil during dynamic joint activities. Ligaments possess a complex microanatomy characterized by an interwoven arrangement of collagen fibrils, primarily type I collagen [16].

Elastin also contributes to the ligaments’ ability to stretch and recoil, accommodating a range of joint movement [17,18]. In the microanatomy, elastin integrates within the stiffer, more rigid collagen network as branching fibrils. The interplay between elastin and collagen in ligaments creates both strength and elasticity. Elasticity is crucial in ligaments associated with joints that undergo a wide range of motion, such as the knee, ankle, and shoulder. Consequently, the higher proportion and strategic arrangement of elastin fibers within ligaments are crucial to their ability to withstand the dynamic mechanical stresses exerted on joints [18].

The surrounding extracellular matrix (ECM) of ligaments contains proteoglycans and glycosaminoglycans, which are critical for supporting tissue hydration, function, and resilience, and providing resistance to compressive forces [19]. The primary cell type within ligaments is fibroblasts, which create and maintain the ECM [15]. Additionally, ligaments have a limited blood supply and a limited innervation, especially at attachment sites, which play a role in healing and proprioception [8,20].

An essential part of the anatomy in ligament repair is the enthesis. The enthesis evenly distributes mechanical stress at the bone-ligament interface, thus preventing injury from localized pressure points. Two distinct types of entheses are found in ligaments: fibrous and fibrocartilaginous. Fibrous entheses directly connect the ligament to the bone, which are common in areas experiencing lower mechanical stress, whereas fibrocartilaginous entheses feature a gradual transition from ligament tissue to bone [21]. The fibrocartilaginous transition involves zones of uncalcified and then calcified fibrocartilage, adeptly managing higher stress loads and providing effective force absorption for major joints—fibrocartilage in tendons and ligaments–an adaptation to compressive load [22]. Entheses are not commonly the site of ligament tears but can be vulnerable to tears if degenerative changes or chronic stress weaken the tissue [21].

### 2.4. Ligament Injury

Ligament injuries commonly affect a wide range of individuals, from athletes to the general population. Ligament injuries, involving the tearing or stretching of ligaments, can occur in various joints, with the knee and ankle being particularly susceptible (see Table 1). In a surgical environment, specific ligaments are more susceptible to damage, with the anterior cruciate ligament (ACL), medial collateral ligament (MCL), lateral collateral ligament (LCL), and ankle ligaments, such as the anterior talofibular ligament (ATFL), being particularly vulnerable [23,24,25]. The biomechanical forces these ligaments endure, along with their distinctive anatomical locations, are associated with an increased incidence of surgical interventions [26].

Several factors are associated with ligament rupture, such as acute high-impact injuries, chronic stress resulting in ligamentous sprain or degeneration, or intrinsic tissue deterioration [27,28]. The tear patterns of ligaments and their recovery potential vary depending on anatomical location, blood supply, and the local environment (e.g., intra-articular vs. extra-articular) [29]. Specific ligaments, like the anterior cruciate ligament (ACL), often rupture in areas of relative vascular deficiency within the knee joint [30]. In contrast, medial collateral ligament (MCL) injuries, which occur in a more vascular region, demonstrate different healing dynamics [31].

Ligament injuries are diverse and complex, encompassing a range of patterns and outcomes. ACL repair surgery, frequently performed in the U.S., primarily addresses mid-substance tears, with older individuals more prone to proximal tears [32]. Posterior Cruciate Ligament (PCL) injuries differ, typically involving complete tears, with their patterns linked to the mechanism of injury [33]. MCL tears often occur with other knee injuries and are marked by medial compartment bone bruises and Stener-like lesions, though specific tear patterns need more study. LCL tears typically present in three forms, with fibular avulsion being the most common [34]. Ankle ligament injuries, such as to the deltoid and ATFL, are complex and often related to broader ankle issues, with the deltoid usually tearing at the medial malleolar attachment [35].

Achilles tendon repairs, particularly for chronic tears, involve tendon transfers, with Flexor Hallucis Longus (FHL) tendon transfers showing low morbidity and a 14.8% complication rate. Patients typically resume daily activities in about 13.7 weeks and sports in 19.6 weeks [36]. P.B. tendon transfers have a lower complication rate and a higher long-term sports return rate and require a slower recovery [37].

## 3. T/L Graft Options

When conservative treatments for thoracolumbar injury prove ineffective, surgical intervention becomes necessary, often involving the use of grafts to repair or stabilize the affected area. These grafts can be sourced from the patient’s own body (autografts), from another human donor (allografts), from a different species (xenografts), or be artificially manufactured (artificial grafts), each with its own set of advantages and potential drawbacks [38,39]. The selection of a graft type is a critical decision in the surgical process, guided by factors such as the nature of the injury, the patient’s health status, and the specific requirements of the spinal repair, aiming to optimize outcomes and minimize complications [40].

While autografts have a high level of success, studies have shown that patients still experience donor-site morbidity and functional disability as long-term outcomes [41,42]. Allografts are harvested from cadaveric human tissues and are often preferred over autografts in select cases due to their shorter operative time, reduced donor site morbidity, and decreased post-operative pain. Although the biomechanical properties of allografts vary depending on the tissue source and processing methods, commonly used allografts can often match or even exceed the properties of native tissue. Furthermore, allografts allow surgeons to tailor the graft size to each patient’s unique anatomy [43]. While they may be favored in specific scenarios, allografts come with constraints such as limited tissue availability, the potential for immune rejection, and the risk of pathogen transmission [44].

Additionally, sterilization and storage techniques can modify the biomechanical properties of allografts. Electron beam and gamma radiation are frequently employed methods for sterilizing allografts but may decrease graft load-to-failure [45,46]. The choice of preservation technique can also alter the properties of grafts. Fresh-freezing tendons can lead to a decrease in load-to-failure, ultimate strain, and ultimate deformation [47]. Alternative preservation techniques, such as glycerolization and lyophilization, demonstrated a significant decrease in load-to-failure. Allografts are a viable option for tendon/ligament reconstruction, but factors like patient age and activity level significantly determine whether an allograft or autograft is preferred [48].

The use of xenografts, particularly those harvested from bovine and porcine tissues, has become a prevalent practice in repairing and reconstructing human ligaments and tendons. Porcine-derived products, in particular, have seen widespread use due to their close resemblance to human tissue in structure and biomechanical properties [49]. This similarity facilitates better integration and functionality of the graft within the human body, making porcine grafts a preferred choice for many surgical interventions. The prevalence of porcine products can also be attributed to the rigorous processing techniques that reduce the risk of immune responses or disease transmission, ensuring their safety and effectiveness in clinical use. Moreover, due to the abundant available tissue, xenografts could provide an extensive supply of graft material without other graft materials [49,50]. Early clinical studies by Dahlstedt et al. used bovine xenografts for ACL reconstruction in 28 patients; within three years, 18 of 28 grafts failed [51]. Recently, a phase 1 clinical trial by Galili et al. using porcine bone-patellar-bone (BPB) xenografts in highly athletic patients [52]. At 24-month follow-up, they reported no significant performance differences between porcine BPB xenograft and cadaveric allograft if non-evaluable subjects were excluded [53]. Despite the benefits, using porcine and other xenografts remains an area of ethical debate and scientific research. Xenografts exhibit drawbacks, including immunogenicity, the risk of disease transmission, and the possibility of graft failure [49,50].

### 3.1. Artificial

The term artificial here is used to describe synthetic polymer-based constructs rather than biological allografts or xenografts. Materials such as polyethylene terephthalate (PET, Dacron), polytetrafluoroethylene (PTFE, Gore-Tex), and polyurethane have been developed into woven or braided scaffolds that aim to reproduce tendon-like tensile behavior. While these polymers provide high initial strength (typically 50–150 MPa) and resistance to creep, they lack biological remodeling and often fail through fatigue or poor integration. To avoid confusion, biologic matrices such as GraftJacket, Restore, and CuffPatch—derived from decellularized dermis—are better classified as biologic or xenograft scaffolds rather than truly artificial materials.

The Ligament Advanced Reinforcement System (LARS) and the Kennedy ligament augmentation device (Kennedy LAD) are two examples of artificial constructs used in ACL reconstruction surgeries. LARS is fabricated from polyethylene terephthalate (PET) fibers, whereas the Kennedy LAD is made from braided polypropylene [54,55,56]. LARS demonstrates a tensile force of 998 ± 148 N (≈50 ± 7 MPa) [55]. An example of an allograft biologic is GraftJacket™, derived from a human dermal matrix, while xenograft biologics include Restore™ and Permacol™, both derived from the porcine small intestine, and Cuff-Patch™, derived from bovine dermal matrix. GraftJacket™ exhibits a tensile strength of 157–229 N (≈8–11 MPa), whereas Restore™ and Permacol™ report tensile strengths of 38 N (≈2 MPa) and 128 N (≈6 MPa), respectively, and Cuff-Patch™ has a tensile strength of 32 N (≈1.5 MPa) [57,58].

Anterior cruciate ligament (ACL) reconstruction commonly relies on autografts, allografts, or xenografts, with patellar and hamstring tendons among the most frequently used grafts. A meta-analysis of more than 47,000 ACL reconstructions reported comparable rupture rates for bone–patellar, tendon-bone, and hamstring grafts. Notably, younger patients demonstrated higher reinjury rates with allografts than autograft revision. Functional outcomes, such as those measured by the Lysholm scale, generally show significant improvements postoperatively [54].

A systematic review of 69 studies found that 81% of patients returned to some form of athletic activity following ACL reconstruction, with 65% achieving their pre-injury level and 55% of high-level athletes successfully resuming competition [55]. However, these figures remain below 90% of patients who attain near-normal knee function. Additionally, early return to sport is correlated with elevated reinjury rates [56].

### 3.2. Conventional Grafts

Conventional reconstruction relies primarily on autografts and allografts, whose selection depends on the injured structure. For example, ACL repair commonly uses hamstring or bone–patellar tendon–bone autografts, whereas PCL reconstructions may employ tibialis anterior or Achilles allografts. MCL and LCL injuries are typically augmented rather than fully replaced but may use semitendinosus or gracilis autografts when reconstruction is required. Each graft type differs in strength (≈800–2500 N (≈40–125 MPa)) and stiffness, influencing postoperative stability and rehabilitation. Although biologically compatible, these grafts carry drawbacks such as donor-site morbidity and delayed remodeling.

Anterior cruciate ligament (ACL) repair surgery is one of the most common surgeries in the United States, with over 120,000 reconstructions performed each year [54]. In their examination of ACL tear patterns, Vermeijden et al. identified that most patients experience mid-substance tears, with proximal tears being more prevalent in older individuals. Anatomic risk factors were not associated with influencing tear location. A link was noted between medial compartment bone contusions and mid-substance tears, which suggests a mechanism of injury for ACL tears beyond anatomic location [57].

For posterior cruciate ligament (PCL) tears, data on specific rupture sites remain limited. However, Sonin et al. reported that PCL injury patterns differ from those of the anterior cruciate ligament (ACL), correlating closely with the mechanism of injury; complete tears were most common, followed by partial tears and bony avulsions [58]. When evaluating medial collateral ligament (MCL) tear patterns, these injuries often occur alongside other knee pathologies and medial compartment bone bruises, particularly at the anteromedial femoral condyle [59]. Furthermore, Stener-like lesions of the superficial MCL (sMCL)—a distal tear with interposition of bone or soft tissue between the ligament and its tibial attachment—are frequently observed in multi-ligamentous knee injuries [60]. Nonetheless, further research is required to clarify the precise tear patterns within the MCL.

Lastly, when discussing LCL tears of the knee, Kahan et al. discussed that there are three distinct patterns of injury, with fibular avulsion being the most common zone of injury (65%), followed by avulsion (20%) and mid-substance tear (15%) [61,62]. Regarding the ankle, such as the deltoid and ATFL ligaments, injury patterns were convoluted and regularly associated with contributing ankle pathology and pattern of injury. The medial malleolar attachment is the most common tear site of the superficial deltoid ligament [63]. Meanwhile, the most common inversion mechanism of injury with supination and plantar flexion of the foot and external rotation of the tibia contributes to the avulsion of the ATFL off the talus [64].

Functional outcomes of Achilles tendon repair also show promise for longevity. Achilles tendon ruptures are often not diagnosed within six weeks, classifying them as a chronic tear and thus making them a surgical candidate for transfer of the flexor hallucis longus (FHL) or peroneus brevis (P.B.). The FHL tendon transfer shows low morbidity, even in sports that require strong push-offs or balance, such as running [65]. The complication rate with a FHL transfer is 14.8% compared to 7% in those with a P.B. transfer. Regarding functionality, patients could return to daily activity in 13.7 weeks and sport in 19.6 weeks [66]. Maffulli et al. reported a slower return to sport in P.B. tendon transfer patients compared to FHL transfer patients, but a higher percentage of P.B. transfer patients eventually returned to sport compared to FHL transfer patients [66].

### 3.3. Silk Grafts

Silk is commonly viewed as a promising material for tendon and ligament tissue engineering, offering unique advantages that address the complex needs of these tissues [67,68]. Recombinant spider silk protein has emerged as a promising biomaterial for augmented tendon repair due to its high tensile strength and elasticity—applications of silk-based biomaterials in biomedicine and biotechnology [69]. By mimicking the organizational structure and mechanical properties of native tendons, engineered spider silk constructs could promote natural integration when surgically implanted at sites of enthesis injury, restoring intrinsic force transmission pathways along the tendon.

Silk’s distinctive combination of biocompatibility, mechanical properties, and versatility in modification allow silk to be a strong candidate for developing constructs that mimic the native structure and function of tendons and ligaments [70,71]. The exploration of silk for T/L replacement highlights the potential to improve patient outcomes by addressing the limitations of current synthetic and biological materials used in T/L engineering. Silk’s compatibility ensures that the engineered tissue can support the body’s movements without premature structural failure while facilitating effective biointegration [71,72].

One of the most important properties of silk for tendon and ligament (T/L) replacement is its tensile strength. Native tendons typically have a tensile strength of 50–100 MPa, while ligaments range from 40 to 80 MPa [73]. See Figure 4 for biomechanical properties of commonly injured Tendons and Ligaments. Silk, with its higher tensile strength, more closely replicates the mechanical demands required for the dynamic and structural functions of T/L. Silkworm silk, for instance, has a tensile strength of 300–500 MPa, [68] making it a strong candidate for T/L replacement. However, it is still outmatched by spider silk, exhibiting exceptional tensile strength ranging from 1000 to 1300 MPa [74].

For clarity, all mechanical values are reported here in MPa to allow direct comparison with natural tendon and ligament tissues, which typically exhibit ultimate tensile strengths of 50–150 MPa. While spider silk demonstrates exceptional properties in this range, its clinical translation is constrained by limited natural availability. True spider silk can only be harvested in milligram quantities, making it impractical for large-scale use. Current research therefore focuses on recombinant spider-silk-like proteins, which reproduce partial sequence motifs but often yield lower mechanical performance and significantly higher production costs. In contrast, silkworm fibroin is abundant and can be processed through braiding, weaving, or knitting to enhance fiber alignment, anisotropy, and tensile strength—features essential for tendon–ligament applications.

Elasticity is another requirement of tendons and ligaments (T/L) that is essential for their functionality. Elasticity enables T/L to absorb shock, facilitate movement, and maintain stability within the musculoskeletal system. Typically, the Young’s modulus for tendons ranges from 500 to 1500 MPa and from ~200 to 600 MPa for ligaments, reflecting their capacity to stretch and return to their original shape. This elasticity can vary significantly with age, activity level, and exposure to injury [73]. Biomaterial research aims to replicate this elastic behavior in T/L replacements. Both silkworm and spider silk show promise because of their notable elasticity, which can closely resemble or even surpass that of native T/L [68]. Silkworm silk demonstrates an elastic modulus of approximately 6000–12,000 MPa, whereas spider silk can reach up to 15,000 MPa, presenting exciting opportunities to enhance the resilience and durability of T/L prosthetics [68,74]. Spider silk can also stretch 30–40% before failing, emphasizing its potential to replicate the flexibility and resilience of human tendons and ligaments. This exceptional elasticity and high tensile strength position spider silk as a viable option for functional tissue replacements, potentially offering advantages over natural T/L in specific applications [72].

The adaptability of silk for T/L extends to its surface properties, which can be finely tuned to foster cell behavior critical for tissue repair.

By employing methods like the incorporation of cell-attachment sequences (for instance, RGD motifs), silk’s surface could become a fertile ground for the adhesion, expansion, and specialization of tendon and ligament cells. This modification enhances the material’s biological compatibility, creating an optimal environment for tissue regeneration to encourage cells to anchor securely, multiply, and develop into mature, functional tissue components. Such capabilities are pivotal in reconstructive therapies, offering a promising pathway toward restoring tendon and ligament functionality through engineered tissue integration [75,76].

Biocompatibility is paramount in reducing implant rejection risk, promoting more seamless tissue integration and regeneration [68]. Silk is naturally non-cytotoxic, resorbable, and minimally immunogenic, making it an attractive option for implantation [74]. This characteristic is especially beneficial in tissue engineering and regenerative medicine, where immune tolerance is essential for successful outcomes [68]. In contrast to some synthetic biomaterials that can trigger strong inflammatory or fibrotic responses, silk’s bio-friendly profile helps ensure it is perceived as benign by the body’s immune system [74]. Additionally, silk’s inherent resorbability obviates the need for surgical removal once the tissue has sufficiently repaired or regenerated. Its degradation rate can be tailored through various processing methods, allowing the scaffold to remain structurally supportive as needed before naturally breaking down [68,74]. This combination of minimal immune response and tunable biodegradation makes silk an ideal candidate for advanced biomedical devices, enhancing tissue repair while minimizing patient discomfort.

Silk’s versatility extends to tailoring its mechanical properties through fiber design and surface modification. By adjusting the spinning conditions, silk fiber composition, and scaffold architecture, researchers can create scaffolds that closely emulate these mechanical and structural characteristics of T/L (see Figure 2). This tuneability allows for the design of personalized implants that meet the specific needs of individual patients and injury types [68].

Silk-based biomaterials offer a promising approach to addressing the complex extracellular matrix (ECM) properties of ligaments, which are essential for structural integrity, mechanical resilience, and biological function [22,75]. Native ligament ECM is composed primarily of type I collagen, elastin, proteoglycans, and glycosaminoglycans, which together provide tensile strength, elasticity, and hydration to withstand multi-directional mechanical loads [22]. The hierarchical organization of these components is critical for ligament function and must be replicated in tissue-engineered constructs. Silk fibroin, derived from silkworms or recombinant sources, possesses tunable mechanical properties and structural compatibility that allow it to mimic the collagen-rich ECM of ligaments while offering superior biocompatibility and controlled degradation [68].

While we note that silk’s degradation can be tuned by processing, this has direct mechanical consequences under physiologic loading [76,77,78]. In vivo hydrolytic/enzymatic loss of amorphous domains and β-sheet reorganization reduce fiber cross-section and alter crystallinity, which lowers modulus and UTS and increases creep [79]. Under cyclic (fatigue) loading, these time-dependent changes manifest as progressive elongation, hysteresis growth, and earlier crack initiation at fiber–bundle junctions—i.e., mechanical degradation of the construct [80]. Thus, “controlled degradation” must be balanced against retention of fatigue strength across the rehabilitation window; processing variables that slow mass loss (e.g., higher β-sheet content, tighter draw, denser braid/knit) generally improve cyclic durability but may delay cellular infiltration [78,81]. Designing silk T/L scaffolds therefore requires co-optimization of degradation kinetics and fatigue life rather than static strength alone [80].

Furthermore, silk scaffolds can be engineered with aligned nanofibrous structures to promote collagen deposition and facilitate cell-mediated ligament regeneration [81,82]. RGD functionalization of silk is technically well-established and achievable through multiple routes [83]. Short RGD peptides can be covalently grafted onto silk fibroin via carbodiimide (EDC/NHS) coupling to carboxylates or via tyrosine-targeted diazonium/azo chemistry [83]; RGD can also be presented within enzymatically cross-linked silk/silk-gelatin hydrogels (e.g., horseradish peroxidase systems) with tunable stiffness and bioactivity [80]. Alternatively, RGD motifs can be encoded genetically into recombinant silk-like proteins to yield uniform, sequence-defined Presentation along the backbone [79,84]. Across formats (films, fibers, hydrogels), RGD-bearing silk substrates increase tendon-cell adhesion and promote tenogenic differentiation and ECM deposition, improving early integration at the scaffold surface [79,80,81,82,83,84,85].

In practice for T/L constructs, we anticipate using braided/knitted fibroin fibers followed by low-degree surface activation (e.g., mild EDC/NHS in MES buffer, pH ~6.0) to immobilize RGD at surface densities shown to enhance integrin-mediated adhesion without over-softening the fiber coating—while preserving fiber strength; for hydrogel interphases (e.g., at the enthesis), HRP-mediated cross-linking of silk–gelatin with RGD provides a conformal, cell-interactive layer that does not impede load transfer [80,81,82,85]. Additionally, silk’s ability to be functionalized with bioactive moeities, such as cell adhesion peptides and growth factors, further improves its capacity to guide ligament healing and regeneration [78]. Unlike synthetic polymers that may provoke inflammatory responses or lack bioactivity, silk fibroin naturally integrates with native ECM components and gradually resorbs without eliciting adverse immune reactions [79].

By addressing both the mechanical and biochemical properties of ligament ECM, silk-based scaffolds provide a biomimetic environment that supports ligament regeneration and repair. Advances in biofabrication techniques, such as electrospinning and 3D bioprinting, further enable precise control over silk’s microarchitecture, optimizing its role as an ECM-mimetic scaffold for ligament tissue engineering [80]. Ongoing research continues to refine silk-based constructs to enhance their structural durability and regenerative potential, positioning silk as a viable material for ligament reconstruction and repair.

In conclusion, silk presents a promising approach to T/L replacement. Its mechanical properties, modifiability, non-cytotoxicity, and minimal inflammatory response position it as a valuable material in the quest for effective tissue regeneration solutions.

## 4. Factors Affecting Regeneration

The regeneration of T/L is associated with multifaceted factors, which are critical to understanding enhancing healing outcomes and graft replacement options [81]. These healing factors span biological, mechanical, and environmental domains [38]. Within the biological category, the cellular healing response makeup, including the composition of fibroblasts and stem cells, sets the stage for tissue healing. In the mechanical domain, the physical properties and forces acting on tendons and ligaments, such as tension, compression, and shear stress, play a pivotal role in guiding tissue regeneration and functional recovery, emphasizing the need for biomechanically compatible scaffold designs and rehabilitation protocols. Meanwhile, the environmental domain underscores the importance of the extracellular matrix, biochemical signals, and the overall physiological environment, which collectively influence cellular behavior, matrix remodeling, and the integration of repair tissues with native structures, highlighting the synergy between mechanical cues and biological responses in the healing process [82].

In the biological domain of T/L repair, vascularization amount and quality significantly affect healing by influencing nutrient delivery and waste removal. At the same time, age and genetics determine the intrinsic healing capacity of the individual. Ligaments generally have lower vascular access than connective tissues, impacting their regenerative ability. Reduced vascularization reduces the delivery of essential nutrients and oxygen. As such, ligament injuries often have prolonged recovery times and may necessitate more extensive therapeutic approaches. Understanding this anatomical limitation is essential in developing bioengineered materials to achieve desired outcomes [83].

In addition to vascularization within the biological domain of T/L healing, the cellular response is crucial, characterized by the activity of fibroblasts and stem cells. Fibroblasts synthesize the extracellular matrix and collagen, which are fundamental for tissue strength and repair, while stem cells offer regenerative capabilities, differentiating into various cell types needed for healing. This cellular interplay replaces damaged cells and secretes growth factors that guide the repair process, orchestrating a complex but coordinated effort to restore tissue integrity and function [38]. In addition, Growth factors from the cellular response, including but not limited to the amount and delivery of Vascular Endothelial Growth Factor (VEGF) [84], Platelet-Derived Growth Factor (PDGF), and Transforming Growth Factor-Beta (TGF-β), are essential for regulating cellular activities during repair [2]. Hormonal influences and the body’s inflammatory response further modulate the regeneration process, with the extracellular matrix composition underpinning the structural integrity and function of the repaired tissue.

Mechanical factors introduce a second layer of complexity to T/L healing. The role of mechanical loading is twofold: it serves as a necessary stimulus for tissue remodeling and strength but requires careful modulation to prevent further injury [85]. The alignment and integrity of the tissue are critical for restoring the mechanical properties essential for function. Additionally, the graft type directly impacts the potential for successful regeneration, emphasizing the need for tailored therapeutic approaches based on the specific nature of the injury [86].

Though sometimes less considered, environmental factors, including nutrition and lifestyle choices, are important. Environmental factors are commonly “pure risks” because they have little chance of improving outcomes and performance but come with substantial downside risks [87]. For example, comorbidities such as diabetes or obesity can impair healing by affecting systemic health and local tissue responses. Most notably, medications, including NSAIDs and corticosteroids, also adversely impact regeneration outcomes [87].

Regeneration of T/L is a highly intricate process influenced by biological, mechanical, and environmental factors. Advances in understanding these factors and their interactions will continue to improve therapeutic approaches, enhancing the prospects for successful T/L regeneration. Advances in understanding these factors and the integration of silk with bioactive molecules and growth factors can further enhance regenerative capacity by promoting cellular responses and tissue remodeling. Lastly, optimizing the structural and mechanical properties of silk scaffolds to mimic native tissue closely can significantly influence the outcomes of tendon and ligament repair.

The regenerative outcome of any graft depends on the interplay between biological, mechanical, and environmental factors. Mechanically, scaffolds must sustain early cyclic loading without stress shielding native tissue. Biologically, cellular infiltration and vascularization determine long-term integration. Environmentally, local inflammation, pharmacologic agents, and mechanical rehabilitation protocols modulate remodeling. Relating these factors back to graft choice, silk fibroin’s tunable degradation and surface chemistry make it a promising bridge between the strength of synthetics and the biocompatibility of biologics, though standardized in vivo testing remains necessary.

## 5. Discussion

Here, we discuss the issues associated with using silks for T/L repair. The key issues include the microanatomy of T/L in the manufacturing of silk T/L grafts, the mechanical properties of silk compared to T/L, and the biocompatibility and factors associated with repair through the lens of silk.

While silk may serve in the future as an effective scaffold for T/L repair, it is important to recognize that it does not need to replicate the full complexity of the tissue’s microanatomy. The primary function of silk in this context is to provide a supportive structure that facilitates cellular adhesion, proliferation, and differentiation, thereby guiding the natural regenerative processes. By offering tunable mechanical properties and high biocompatibility, silk scaffolds create a conducive environment for tissue repair without mimicking every intricate detail of native tissue. This approach integrates bioactive molecules and growth factors to enhance regenerative outcomes further, ultimately improving successful tendon and ligament regeneration prospects.

Silk as a biomaterial for tendon and ligament regeneration might evoke a science fiction-like appeal due to its remarkable properties and versatility. Silk’s tunability in terms of strength and elasticity allows for precise customization to meet the unique mechanical and biological requirements of specific tendon and ligament locations. This adaptability is crucial, as different tendons and ligaments in the body endure varying levels of stress and strain, necessitating tailored solutions for optimal repair. By adjusting silk scaffolds’ mechanical strength, porosity, and degradation rate, researchers can create highly specialized constructs that support effective healing and integration with native tissues. This level of customization enhances the therapeutic potential of silk-based interventions and pushes the boundaries of regenerative medicine, making science fiction-like advancements a tangible reality.

Although silk possesses functional biomechanical characteristics for tendon and ligament replacement, its translation into a clinically viable graft remains unrealistic at this time. The primary limitations include high production costs, difficulty achieving controlled spinning and post-processing, and the absence of scalable manufacturing methods capable of producing consistent, load-bearing fibers. Achieving reproducible fiber alignment, β-sheet crystallinity, and mechanical uniformity at kilogram scales continues to challenge current production systems. This is evidenced by the inability of companies such as Bolt Threads and Nexia Biotechnologies to successfully upscale production. These examples highlight the persistent economic and technical challenges in transitioning silk from a promising biomaterial to a practical, clinically deployable graft. Furthermore, the lack of standardized quality control protocols, regulatory guidance, and validated large-animal or human clinical data restricts progress toward approval. Consequently, despite its biomechanical suitability and biological promise, silk should presently be regarded as an experimental biomaterial scaffold rather than a realistic graft option for widespread clinical use.

## 6. Conclusions

In conclusion, silk stands out as a promising scaffold for tendon and ligament regeneration due to its unique properties. Its high tensile strength and elasticity closely mimic the mechanical characteristics required for these tissues, ensuring durability and flexibility. Silk’s biocompatibility and non-cytotoxic nature minimize the risk of immune rejection and support cellular adhesion, proliferation, and differentiation. Additionally, silk’s tunable mechanical properties and surface modifiability allow for customization to meet the precise requirements of different tendon and ligament locations. These attributes and silk’s ability to integrate bioactive molecules and growth factors make it an excellent candidate for enhancing tissue repair and improving patient outcomes.

Several research areas need focused attention to overcome the limitations of using silk as a scaffold for T/L regeneration. First, it is crucial to improve the tunability of the mechanical properties of silk scaffolds to ensure they match the specific tensile strength and elasticity required for various applications. Improving the tunability of silk involves investigating different silk sequences and spinning techniques. Second, enhancing the biocompatibility and biointegration of silk requires studying its interactions with various cell types and tissues and modifying its surface properties to promote better cell adhesion, proliferation, and differentiation. Long-term in vivo studies are essential to assess the immune response and integration with native tissues. Third, incorporating bioactive molecules such as growth factors and peptides into silk scaffolds can enhance their regenerative capacity by promoting specific cellular responses and improving tissue remodeling. Fourth, exploring the optimal structural and architectural design of silk scaffolds to mimic the native microanatomy of tendons and ligaments is critical. Optimizing silk design includes utilizing advanced fabrication techniques like electrospinning, 3D printing, and bioprinting. Fifth, understanding and controlling silk scaffolds’ degradation and resorption rates are necessary to ensure they provide temporary scaffolds for the healing process and gradually degrade as native tissue regenerates. Finally, conducting extensive preclinical studies followed by well-designed clinical trials is vital to validate silk scaffolds’ safety, efficacy, and long-term outcomes in tendon and ligament repair. Addressing these research areas will help realize the full potential of silk as a practical and versatile material for tendon and ligament repair.

## Figures and Tables

**Figure 1 polymers-17-03052-f001:**
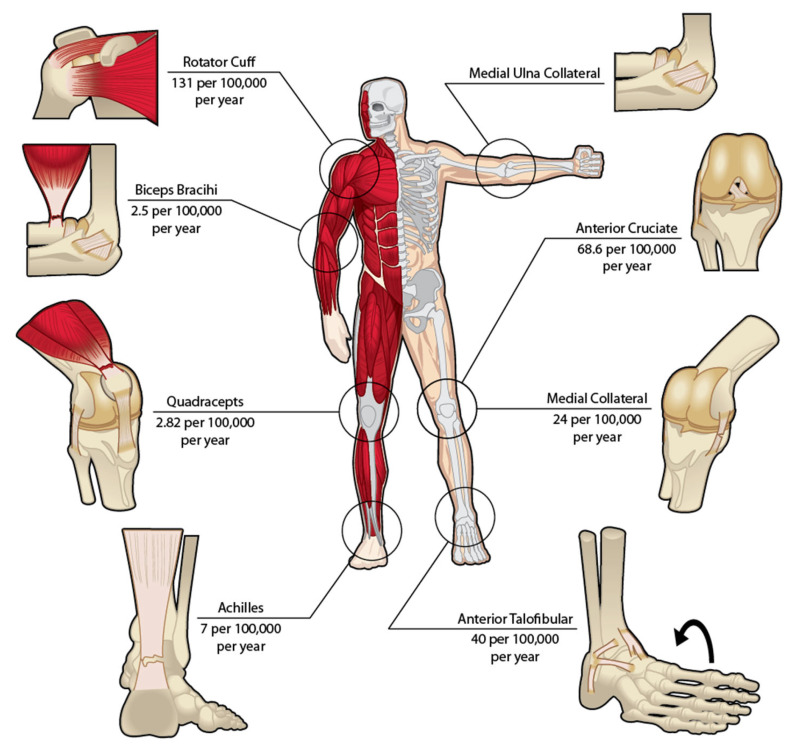
Incidence rates per 100,000 of commonly injured Tendons (**Left**) and Ligaments (**Right**).

**Figure 2 polymers-17-03052-f002:**
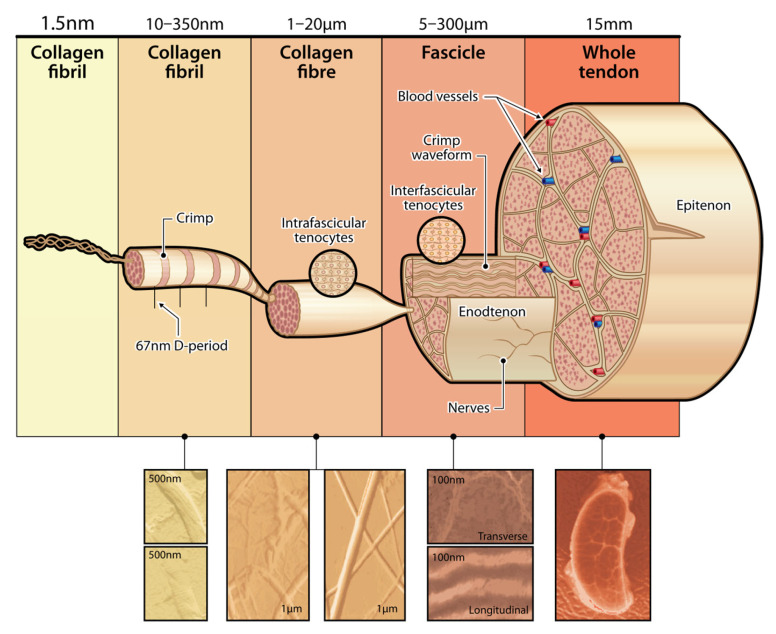
Hierarchical structure of tendon, from collagen fibrils to the whole tendon, demonstrating how organization supports high load resistance.

**Figure 3 polymers-17-03052-f003:**
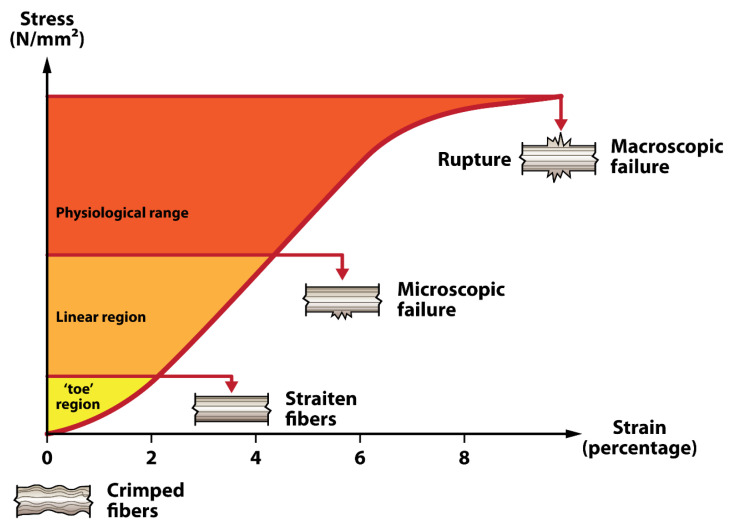
Tendon stress–strain curve illustrating the different phases of tendon deformation. The curve consists of three primary regions: the toe region, where crimped fibers begin to straighten; the linear region, where fibers align and tensile stiffness increases; and the physiological range, which includes both the toe and linear regions and represents normal functional loading. Beyond this range, microscopic failure occurs as individual fibers begin to rupture, followed by macroscopic failure and complete tendon rupture at higher strains. The diagram visually represents these phases with color coding, highlighting the progressive nature of tendon damage under increasing strain.

**Figure 4 polymers-17-03052-f004:**
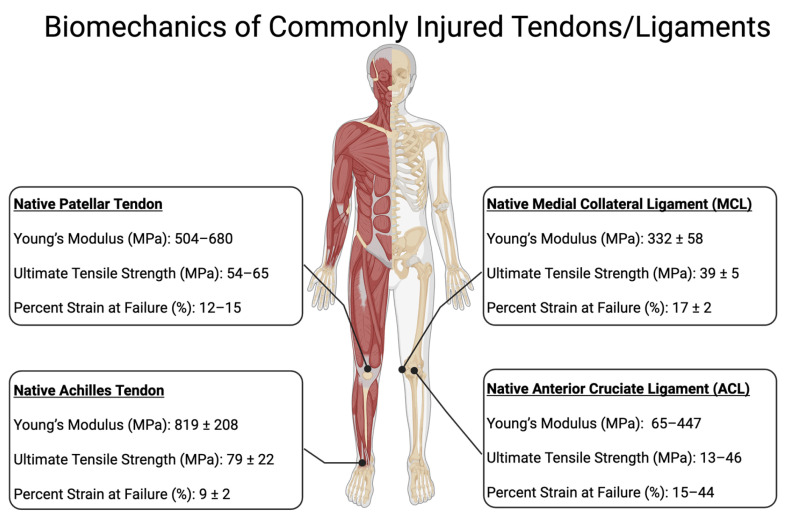
Native biomechanical properties of commonly injured tendons and ligaments. Comparison of the native mechanical properties of the patellar tendon, Achilles tendon, anterior cruciate ligament (ACL), and medial collateral ligament (MCL). Young’s modulus represents the tendon or ligament’s stiffness and its ability to resist deformation under load. Ultimate tensile strength (UTS) indicates the maximum stress the tissue can withstand before failure. Percent strain at failure reflects the tissue’s capacity to elongate before rupture. These parameters provide insight into the functional mechanical behavior of these tissues in the human body.

**Table 1 polymers-17-03052-t001:** Mechanisms of injury and common causes for ligament and tendon tears. The table outlines typical patterns of trauma, both contact and non-contact, as well as sports and activities most frequently associated with each injury.

Ligament/Tendon Injury	Mechanism of Injury	Common Causes
Anterior Cruciate Ligament (ACL) Tear	Non-contact, multi-planar movement involving sudden deceleration, an aggressive pivot or cut, and/or awkward landing from a jump	Common in sports like soccer, football, and basketball
Medial Collateral Ligament (MCL)/Lateral Collateral Ligament (LCL) Tear	Direct Impact: Direct blow to the Outside (MCL) or Inside (LCL) of the knee Non-contact force: Sudden, forceful movement that twists the knee	Direct Impact: contact sports such as hockey or football Non-contact: skier’s foot getting caught or a soccer player cutting and pivoting suddenly
Posterior Cruciate Ligament (PCL) Tear	Direct Impact: Direct blow to the front of the knee Non-contact force: Hyperflexion of the knee, or Sudden forceful movement that twists the knee	Direct Impact: Dashboard injury or falling on a bent knee Non-contact: a misstep causing knee hyperflexion, soccer player cutting and pivoting suddenly
Ulnar Collateral Ligament (UCL) Injury	Repetitive stress from throwing or overhead activities, leading to gradual wear and tear, or a sudden traumatic event like a fall on an outstretched arm	Most common in Baseball Pitchers
Anterior Talofibular Ligament (ATFL) (ankle)	Sudden twisting or inversion of the ankle commonly from a misstep or quick pivot motion	Common in sports like basketball, football, soccer
Achilles Tendon Tear	Sudden, forceful plantarflexion (e.g., pushing off the foot while the knee is extended) or Sudden, unexpected dorsiflexion (e.g., landing from a jump or fall)	Common in sports like gymnastics, basketball, and soccer
Patellar Tendon Tear	Landing from a jump in sports (e.g., basketball, volleyball): The tendon may tear as the quadriceps contract to control landing with the knee flexed.	Common in sports like, basketball, and volleyball
Biceps Tendon Tear	Sudden, forceful resistance against biceps contraction	Lifting heavy weight with arms Dog suddenly and forcefully pulling on leash with outstretched arm

## Data Availability

No new data were created or analyzed in this study.

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
