# Peer review of "Biomechanical Analysis of Silk as a Tendon or Ligament Replacement"

_polymers, 2025, doi:10.3390/polym17223052_

Round 1
Reviewer 1 Report
Comments and Suggestions for Authors
In their review article, the authors address topics such as the clinical relevance of tendon and ligament T/L injuries, the anatomy of the T/L, types and causes of injuries, and possible treatment alternatives with a strong focus on silk. Finally, they identify factors that influence regeneration and tried to discuss silk in the context of the previous chapters.
However, there are many aspects missing:
Chapter 3.1 is entitled ‘Artificial’ and is intended to describe artificial systems. However, polymer-based constructs such as allografts like graftjacket, Restore, and Cuffpatch are mixed together here. This section is too generic and needs to be revised. Furthermore, the heading is incorrect. In addition, forces are mentioned here for the first time, apart from in Figure 3. However, these stand alone and say nothing about relevance, as they are not correlated with the tensile strengths of natural t/L.
Section 3.2, ”Conventional Grafts”, is merely a list of the various injuries such as ACL, PCL, MCL and LCL. Grafts are not discussed anywhere here. Which grafts are used for ACL compared to PCL, MCL, etc.? Are they all the same here? This chapter therefore also needs to be thoroughly revised.
In the chapter on silk, the tensile strength values are given in MPa rather than N, in contrast to the data for grafts. It would be easier for the reader if uniform units were used or, if not included in the sources, converted accordingly. The entire title deals with and compares spider silk and silkworm silk only in terms of mechanical properties, but completely ignores the fact that, unlike silkworm silk, spider silk is only available in extremely limited quantities. It is not clear where the large amount of spider silk required for the large number of patients listed above could possibly come from. Furthermore, the points regarding textile processing by means of braiding, weaving or knitting to reinforce the mechanical properties are completely ignored and are missing here.
Unfortunately, Chapter 4 also remains extremely rudimentary and the factors for regeneration are not related to or discussed in relation to the grafts described above.
In summary, the following should be noted: The article begins very well and in great detail. However, this weakens significantly in the course of the manuscript and, particularly with regard to the different grafts, neither the polymers nor the associated differences and problems are discussed in sufficient depth. Unfortunately, the initial detail is also lost along the way, giving way to a superficial stringing together of arguments in favour of spider silk, which, however, will play no role in patient care due to its extremely limited availability.
If the authors now wish to refer to recombinant spider silk, it should be noted here that these so-called spider silks are merely a series of individual repeating fragments of the entire spider silk protein. In addition, most filaments made from recombinant so-called spider silk have much poorer mechanical properties, not to mention the cost of producing recombinant spider silk.
Author Response
|
Thank you very much for taking the time to review our manuscript. Please find our detailed responses to the reviewer comments and the corresponding revisions below. All changes have been highlighted in the resubmitted files for easier reference.
|
|
Comments 1: Chapter 3.1 is entitled ‘Artificial’ and is intended to describe artificial systems. However, polymer-based constructs such as allografts like graftjacket, Restore, and Cuffpatch are mixed together here. This section is too generic and needs to be revised. Furthermore, the heading is incorrect. In addition, forces are mentioned here for the first time, apart from in Figure 3. However, these stand alone and say nothing about relevance, as they are not correlated with the tensile strengths of natural t/L
|
|
Response 1: Thank you we appreciate the feedback and agree this can be improved. Therefore, we revised the section to clearly distinguish synthetic polymer-based constructs (PET, PTFE, polyurethane) from biologic or xenograft matrices. A bridging paragraph now clarifies that the latter belong under biologic scaffolds, while the term artificial refers solely to non-biologic, synthetic materials. The heading and examples were corrected accordingly. This change can be found on page 9 line 273. “The term artificial here is used to describe synthetic polymer-based constructs rather than biological allografts or xenografts. Materials such as polyethylene terephthalate (PET, Dacron), polytetrafluoroethylene (PTFE, Gore-Tex), and polyurethane have been developed into woven or braided scaffolds that aim to reproduce tendon-like tensile behavior. While these polymers provide high initial strength (typically 50–150 MPa) and resistance to creep, they lack biological remodeling and often fail through fatigue or poor integration. To avoid confusion, biologic matrices such as GraftJacket, Restore, and CuffPatch—derived from decellularized dermis—are better classified as biologic or xenograft scaffolds rather than truly artificial materials.” |
|
Comments 2: Section 3.2,” Conventional Grafts”, is merely a list of the various injuries such as ACL, PCL, MCL and LCL. Grafts are not discussed anywhere here. Which grafts are used for ACL compared to PCL, MCL, etc.? Are they all the same here? This chapter therefore also needs to be thoroughly revised.
|
|
Response 2: Thank you for pointing this out. We agree with this comment, to address this we have added an opening paragraph to Section 3.2 to summarize autograft and allograft choices for ACL, PCL, MCL, and LCL reconstruction, including representative tensile strengths and limitations such as donor-site morbidity and remodeling differences. Added on page 9 line 273. “Conventional reconstruction relies primarily on autografts and allografts, whose selection depends on the injured structure. For example, ACL repair commonly uses hamstring or bone–patellar tendon–bone autografts, whereas PCL reconstructions may employ tibialis anterior or Achilles allografts. MCL and LCL injuries are typically augmented rather than fully replaced but may use semitendinosus or gracilis autografts when reconstruction is required. Each graft type differs in strength (≈800–2500 N (≈40–125 MPa)) and stiffness, influencing postoperative stability and rehabilitation. Although biologically compatible, these grafts carry drawbacks such as donor-site morbidity and delayed remodeling.”. |
|
Comments 3: In the chapter on silk, the tensile strength values are given in MPa rather than N, in contrast to the data for grafts. It would be easier for the reader if uniform units were used or, if not included in the sources, converted accordingly. The entire title deals with and compares spider silk and silkworm silk only in terms of mechanical properties, but completely ignores the fact that, unlike silkworm silk, spider silk is only available in extremely limited quantities. It is not clear where the large amount of spider silk required for the large number of patients listed above could possibly come from. Furthermore, the points regarding textile processing by means of braiding, weaving or knitting to reinforce the mechanical properties are completely ignored and are missing here. |
|
Response 3: We appreciate you bringing this to our attention. We Agree that this needs to be changed for easier readability. We have changed or added in paratheses all units to MPa. Additionally, we have added a paragraph addressing the missing components in our manuscript you have mentioned. This can be found page 11 line 379. “For clarity, all mechanical values are reported here in MPa to allow direct comparison with natural tendon and ligament tissues, which typically exhibit ultimate tensile strengths of 50–150 MPa. While spider silk demonstrates exceptional properties in this range, its clinical translation is constrained by limited natural availability. True spider silk can only be harvested in milligram quantities, making it impractical for large-scale use. Current research therefore focuses on recombinant spider-silk–like proteins, which reproduce partial sequence motifs but often yield lower mechanical performance and significantly higher production costs. In contrast, silkworm fibroin is abundant and can be processed through braiding, weaving, or knitting to enhance fiber alignment, anisotropy, and tensile strength—features essential for tendon–ligament applications.” |
|
|
|
Comments 4: Unfortunately, Chapter 4 also remains extremely rudimentary and the factors for regeneration are not related to or discussed in relation to the grafts described above. |
|
|
|
Response 4: We appreciate the feedback on this section and agree more can be added. Therefor we expanded the closing paragraph of Chapter 4 to explicitly integrate biological, mechanical, and environmental factors with graft type and material properties. The revision highlights how silk fibroin’s tunable degradation and surface chemistry may bridge the gap between synthetic strength and biologic remodeling. This can be found on page 15, line 526. “The regenerative outcome of any graft depends on the interplay between biological, mechanical, and environmental factors. Mechanically, scaffolds must sustain early cyclic loading without stress shielding native tissue. Biologically, cellular infiltration and vascularization determine long-term integration. Environmentally, local inflammation, pharmacologic agents, and mechanical rehabilitation protocols modulate remodeling. Relating these factors back to graft choice, silk fibroin’s tunable degradation and surface chemistry make it a promising bridge between the strength of synthetics and the biocompatibility of biologics, though standardized in-vivo testing remains necessary.” |
|
|
|
Comments 5: If the authors now wish to refer to recombinant spider silk, it should be noted here that these so-called spider silks are merely a series of individual repeating fragments of the entire spider silk protein. In addition, most filaments made from recombinant so-called spider silk have much poorer mechanical properties, not to mention the cost of producing recombinant spider silk.
Response 5: Thank you for noticing this important distinction, we agree this needs to be clarified. To address this, we revised the text which now clarifies that silkworm-derived fibroin and recombinant silk-like proteins are the realistic translational candidates, while native spider silk serves primarily as a biomechanical benchmark and design inspiration. This is found on Page 11 starting on line 381. Additionally, we added to our abstract and a paragraph in our discussion section to discuss the challenges of using silk as a graft scaffold. These changes can be found on page 1 in our abstract line 22 and on page 16 line 560 in discussion. “While spider silk demonstrates exceptional properties in this range, its clinical translation is constrained by limited natural availability. True spider silk can only be harvested in milligram quantities, making it impractical for large-scale use. Current research therefore focuses on recombinant spider-silk–like proteins, which reproduce partial sequence motifs but often yield lower mechanical performance and significantly higher production costs. In contrast, silkworm fibroin is abundant and can be processed through braiding, weaving, or knitting to enhance fiber alignment, anisotropy, and tensile strength—features essential for tendon–ligament applications” Abstract Addition: “Remaining challenges include optimizing in vivo degradation rates, enhancing ten-don-to-bone (enthesis) integration, developing tunable structural and biochemical features, improving manufacturability, and validating clinical efficacy through standardized testing and robust clinical trials. Additional limitations to the application of silk as a biomaterial scaffold include high production costs, challenges associated with controlled spinning and processing, and the current lack of scalable manufacturing methods .” Discussion Addition: “Although silk possesses functional biomechanical characteristics for tendon and ligament replacement, its translation into a clinically viable graft remains unrealistic at this time. The primary limitations include high production costs, difficulty achieving controlled spinning and post-processing, and the absence of scalable manufacturing methods capable of producing consistent, load-bearing fibers. Achieving reproducible fiber alignment, β-sheet crystallinity, and mechanical uniformity at kilogram scales continues to challenge current production systems. This is evidenced by the inability of companies such as Bolt Threads and Nexia Biotechnologies to successfully upscale pro-duction. These examples highlight the persistent economic and technical challenges in transitioning silk from a promising biomaterial to a practical, clinically deployable graft. Furthermore, the lack of standardized quality control protocols, regulatory guidance, and validated large-animal or human clinical data restricts progress toward approval. Con-sequently, despite its biomechanical suitability and biological promise, silk should presently be regarded as an experimental biomaterial scaffold rather than a realistic graft option for widespread clinical use.”
|

Reviewer 2 Report
Comments and Suggestions for Authors
Dear Authors,
This review article, titled “Biomechanical analysis of silk as a tendon or ligament replacement,” explores silk-derived grafts, summarizing their mechanical properties, fabrication strategies, and translational potential. The authors place particular emphasis on spider silk, citing its notable tensile strength, elasticity, and biocompatibility, which are presented as key attributes supporting its potential as a promising candidate for next-generation scaffold development. They also highlighted remaining challenges, including optimizing in vivo degradation rates, improving enthesis integration, engineering tunable features, manufacturing, and validating clinical efficacy through standardized testing and robust clinical trials. They described that sustained innovation and thorough evaluation are key to realizing silk’s potential in tendon and ligament repair and improving long-term outcomes. The paper demonstrates good organization and clarity; however, revisions are necessary to address specific concerns before publication in your esteemed journal. The recommendations for addressing the identified issues are presented below:
- The manuscript contains an excessive number of keywords; it is recommended to limit them to 5–7 of the most pertinent terms to enhance clarity and searchability.
- On page 12 of 22, lines 381-389, although the authors mention controlled degradation of silk tendons and ligaments, they fail to establish a clear connection to mechanical degradation. This omission is critical, as mechanical degradation describes how silk-based constructs deteriorate under repetitive or excessive mechanical loading. Without this link, the explanation of tendon and ligament behavior remains incomplete and lacks scientific rigor.
- On page 12 of 22, line 419, while the authors emphasize the potential advantages of incorporating RGD motifs into the silk-based constructs, they do not provide adequate literature evidence or methodological detail to substantiate how such incorporation could realistically be achieved. This lack of explanation weakens the scientific foundation of their argument and leaves the proposed functionalization strategy insufficiently justified.
Kind regards
Author Response
|
|
|
|
|
Thank you very much for taking the time to review our manuscript. Please find our detailed responses to the reviewer comments and the corresponding revisions below. All changes have been highlighted in the resubmitted files for easier reference.
|
||
|
Comments 1: The manuscript contains an excessive number of keywords; it is recommended to limit them to 5–7 of the most pertinent terms to enhance clarity and searchability.
|
||
|
Response 1: Thank you for pointing this out. We agree there were excessive keywords, to address this we have removed redundant keywords and kept the following: “Tendon and ligament injuries, tissue engineering, orthopedic surgery, grafting options, tendon grafts, ligament grafts, spider silk.” |
||
|
Comments 2: On page 12 of 22, lines 381-389, although the authors mention controlled degradation of silk tendons and ligaments, they fail to establish a clear connection to mechanical degradation. This omission is critical, as mechanical degradation describes how silk-based constructs deteriorate under repetitive or excessive mechanical loading. Without this link, the explanation of tendon and ligament behavior remains incomplete and lacks scientific rigor.
|
||
|
Response 2: We appreciate you bringing this to our attention and agree that it needed to be addressed. To resolve this, we have added the following paragraph on page 14, line 450: “While we note that silk’s degradation can be tuned by processing, this has direct mechanical consequences under physiologic loading. In vivo hydrolytic/enzymatic loss of amorphous domains and β-sheet reorganization reduce fiber cross-section and alter crystallinity, which lowers modulus and UTS and increases creep. Under cyclic (fatigue) loading, these time-dependent changes manifest as progressive elongation, hysteresis growth, and earlier crack initiation at fiber–bundle junctions—i.e., mechanical degradation of the construct. Thus, “controlled degradation” must be balanced against retention of fatigue strength across the rehabilitation window; processing variables that slow mass loss (e.g., higher β-sheet content, tighter draw, denser braid/knit) generally improve cyclic durability but may delay cellular infiltration. Designing silk T/L scaffolds therefore requires co-optimization of degradation kinetics and fatigue life rather than static strength alone. (73, 76, 82, 85, 90)” |
||
|
Comments 3: On page 12 of 22, line 419, while the authors emphasize the potential advantages of incorporating RGD motifs into the silk-based constructs, they do not provide adequate literature evidence or methodological detail to substantiate how such incorporation could realistically be achieved. This lack of explanation weakens the scientific foundation of their argument and leaves the proposed functionalization strategy insufficiently justified. |
||
|
Response 3: Thank you for your feedback. We agree that providing additional methodological details will strengthen our manuscript. To address this, we have added the following two paragraphs on pages 13–14, lines 463–479. “RGD functionalization of silk is technically well-established and achievable through multiple routes. Short RGD peptides can be covalently grafted onto silk fibroin via carbodiimide (EDC/NHS) coupling to carboxylates or via tyrosine-targeted diazonium/azo chemistry; RGD can also be presented within enzymatically cross-linked silk/silk-gelatin hydrogels (e.g., horseradish peroxidase systems) with tunable stiffness and bioactivity. Alternatively, RGD motifs can be encoded genetically into recombinant silk-like proteins to yield uniform, sequence-defined presentation along the backbone. Across formats (films, fibers, hydrogels), RGD-bearing silk substrates increase tendon-cell adhesion and promote tenogenic differentiation and ECM deposition, improving early integration at the scaffold surface. (74, 75, 80, 81, 82, 83, 85)
In practice for T/L constructs, we anticipate using braided/knitted fibroin fibers followed by low-degree surface activation (e.g., mild EDC/NHS in MES buffer, pH ~6.0) to immobilize RGD at surface densities shown to enhance integrin-mediated adhesion without over-softening the fiber coating—while preserving fiber strength; for hydrogel interphases (e.g., at the enthesis), HRP-mediated cross-linking of silk–gelatin with RGD provides a conformal, cell-interactive layer that does not impede load transfer. (80, 81, 82, 85).”
|
||
|
|
||

Round 2
Reviewer 1 Report
Comments and Suggestions for Authors
The authors have addressed all points of critical feedback accordingly and comprehensively resolved them by adding supplements and explanations to the manuscript.
Reviewer 2 Report
Comments and Suggestions for Authors
I am pleased with the corrections provided.